# Arterial floating mural thrombi are a characteristic imaging pattern in SARS-CoV-2-related ischemic stroke

**Keshet Pardo**[1,2]*, **Omer Harnof**[2], **Rani Barnea**[1,2], **Jonathan Naftali**[1,2], **Gilad Kenan**[2,3], **Eithan Auriel**[1,2], **Shlomi Peretz**[2,3]

**1** Department of Neurology, Rabin Medical Center – Beilinson Hospital, Petach Tikva, Israel, **2** Sackler School of Medicine, Tel-Aviv University, Tel-Aviv, Israel, **3** Department of Neurology, Shamir Medical Center, Be'er Ya'akov, Israel

* keshetpa@clalit.org.il

**Data Availability Statement:** : All data generated or analyzed during this study are included in this article and its supplementary material files. Further

## Abstract

### Background

Acute ischemic stroke (AIS) is a complication of severe acute respiratory syndrome coronavirus 2 (SARS-CoV-2) infection. We aimed to explore neurovascular imaging patterns in patients with SARS-CoV-2-related AIS.

### Methods

We retrospectively analyzed clinical and radiological data of patients hospitalized with AIS and a positive PCR test for SARS-CoV-2 prior to AIS onset. The control group comprised of AIS patients from a pre-COVID-19 pandemic period matched for gender and age.

### Results

Thirty-five SARS-CoV-2-related stroke patients, and 35 controls were included. Fifty-seven percent of SARS-CoV-2 patients had either mild or asymptomatic disease. A distinctive imaging pattern of floating arterial mural thrombus was detected in 5 patients of the SARS-CoV-2 group. In 4 patients thrombus was attached to a stenotic atherosclerotic plaque in the proximal internal carotid artery. In the 5th patient a cardiac CTA showed multiple floating thrombi in the descending aorta. In the control group, floating thrombus was only detected in one patient. Treatment with dual antiplatelet therapy was associated with thrombus dissolution and good clinical outcome. Patients with floating thrombi had a longer time from SARS-CoV-2 diagnosis to stroke onset (mean 7.4 versus 3.4 days).

### Conclusions

Floating arterial mural thrombi attached to atherosclerotic plaques are unique characteristic source of AIS in SARS-CoV-2 patients. They may lead to ischemic stroke in patients with mild or asymptomatic infection up to 1–2 weeks from SARS-CoV-2 diagnosis. Patients with

enquiries can be directed to the corresponding author.

**Funding:** The author(s) received no specific funding for this work.

**Competing interests:** The authors have declared that no competing interests exist.

embolic AIS and SARS-CoV-2 diagnosis should perform high resolution cranio-cervical vascular imaging to evaluate floating thrombi as a potential embolic source.

## Introduction

Coronavirus disease 2019 (COVID-19), caused by the novel severe acute respiratory syndrome coronavirus 2 (SARS-CoV-2), was first detected in December 2019 and has since become a worldwide pandemic [1]. In addition to the common symptoms of fever, rhinorrhea, cough, and shortness of breath, SARS-CoV-2 may cause a hypercoagulable state associated with acute inflammatory changes. The risk for venous thromboembolism (VTE) is markedly increased. The risk of arterial thrombotic events such as stroke, myocardial infarction, and limb ischemia are also increased, but to a lesser extent than venous thrombosis. Stroke has been reported in 0.5–1.6 percent of hospitalized SARS-CoV-2 patients. Risk factors for arterial thrombosis included older age, male sex, history of coronary artery disease and D-dimer >500 ng/mL on presentation. Arterial thrombotic events were associated with increased mortality [2]. A recent study found that the risk of stroke in adults with SARS-CoV-2 between the ages of 65 to 74 was highest within the first three days after diagnosis compared to 7 days before or 28 after diagnosis (IRR 10.3; 95% CI 9.9–10.8). This excessive risk gradually declined but remained high throughout the 28-day time window [3].

The characteristics and stroke subtypes of SARS-CoV-2-related stroke at large scale have been reported by the Multinational COVID-19 Stroke Study Group [4]. Conclusions from the analysis of 283 ischemic stroke patients were that SARS-CoV-2-associated stroke carries a considerably higher rate of large vessel occlusions, a much lower rate of small vessel occlusion and lacunar infarction, and a considerable number of young stroke when compared with population studies before the pandemic.

In the Rabin medical center (RMC) between 950 and 1,000 patients are admitted with ischemic stroke annually. During the SARS-CoV-2 pandemic the annual incidence of ischemic stroke admissions in RMC rose by about 10%. We aimed to explore the neurovascular imaging data of SARS-CoV-2-related versus non-SARS-CoV-2-related ischemic stroke patients and identify characteristic and possibly unique imaging patterns. Identification of such SARS-CoV-2-related stroke imaging patterns may help to guide etiological evaluation and optimal treatment.

## Methods

### Study design and population

We searched the RMC medical files for patients whose discharge diagnoses included ischemic stroke and were diagnosed with SARS-CoV-2 infection by either an in-hospital or ambulatory polymerase chain reaction (PCR) test from one month before to 7 days after the onset of the ischemic stroke. We included the time periods in which SARS-CoV-2 was routinely screened for in all admitted patients in RMC, between March 2020 and May 2022, representing the surge in Alpha, Delta and Omicron variants in Israel [5]. Severity of the SARS-CoV-2-infection was defined according to the World Health Organization interim guidelines [6].

The control group comprised of non-SARS-CoV-2 associated stroke patients from a pre-COVID-19 pandemic period, matched for gender and age (±5 years) with a case to control ratio of 1:1. The control patients were searched for during a period that predated the COVID-19 pandemic in Israel—between January 2018 and February 2020.

Patients with peri-procedural stroke, hemorrhagic stroke, transient ischemic attack (TIA) or venous thrombosis were excluded from the study.

## Data

A trained medical student (O.H.) retrieved demographics and relevant clinical data including age, sex, vascular risk factors, NIHSS at admission, intravenous thrombolytic (IVT) treatment, endovascular treatment (EVT), pre-stroke modified Rankin Scale (mRS) score, and mortality at 90 days from the medical records. Laboratory data at admission were also retrieved and included WBC, PLT, CRP and fibrinogen. Imaging data included non-contrast head CT (NCCT), CT Angiography (CTA) of the head and neck and head CT perfusion (CTP) performed on admission or upon stroke onset in cases of in-house stroke. Time from SARS-CoV-2 diagnosis to AIS was also recorded.

Stroke evaluation ancillary tests including ECG, echocardiogram, Holter ECG, cervical artery duplex and hypercoagulability battery, along with imaging and clinical data, were examined by an experienced neurologist (K.P.). Stroke subtypes were determined based on the TOAST classification system and strokes were classified as either large artery atherosclerosis, cardioembolic, small artery occlusion, other determined etiology, and undetermined etiology [7].

For patients with SARS-CoV-2 associated stroke, symptoms severity was determined based on the National Institutes of Health COVID-19 Treatment Guidelines Panel as either a-symptomatic, mild, moderate, or severe [8].

The functional outcome (mRS) at 90 days after stroke onset was recorded for all patients either through information from a follow-up visit at a post-stroke outpatient clinic or virtually via a phone call using a simplified mRS questionnaire (smRSq) [9].

The study was approved by the local ethical committee (0190-22-RMC). Due to the retrospective non- interventional design of the study informed consent was not required.

Data was accessed for research purposes on the 1st of June 2022, the data received by authors could not identify individual participants in the study.

## Imaging analysis

Imaging admission data including NCCT, CTA and CTP of both SARS-CoV-2-related stroke patients and the control group were reviewed by an experienced stroke neurologist (S.P.), who was blinded to clinical data and patient group allocation. Neurovascular characteristics were recorded for each patient including the presence of a large vessel occlusion and its location and significant atherosclerosis/stenosis and/or mural thrombi along the vascular tree from the aortic arch to the most distal intracranial arteries visible on CTA. A floating mural thrombus was defined as a filling defect arising proximally from the carotid wall and with circumferential blood flow at its distal aspect.

## Statistical analysis

Statistical analysis was obtained by using IBM SPSS Statistics for Windows, Version 25.0 (IBM Corp., Armonk, NY). Qualitative data was introduced as frequencies and percentages. Pearson's chi-square test was used for comparison. Quantitative data was introduced as either median (IQR) for non-normally distributed data and means ± SD for normally distributed data. T-test was used for comparison of demographic data. Binominal univariant regression was performed to evaluate the association between clinical and stroke characteristics and functional independence at day 90. Ordinal logistic regression analysis, adjusted for age, thrombolytic treatment, and endovascular treatment as used to assess the effect of SARS-CoV-

2-infection on stroke outcome (mRS) at 90 days. P-value $\leq$ 0.05 was considered statistically significant.

## Results

### Baseline characteristics

Thirty-five SARS-CoV-2-related stroke patients and 35 stroke control patients were included in the analysis. The two groups were matched for gender, with 40% females in each group, and for age, with a median of 74 years (IQR; 69, 81) for the SARS-CoV-2 group, and 74 years (IQR; 68, 82) for the control group. There were no statistically significant differences between groups in terms of cardiovascular risk factors, prior stroke or TIA or the presence of an active neoplasm Table 1.

Within the SARS-Cov-2 group 10 patients (28.6%) had asymptomatic disease whereas 10 (28.6%), 6 (17.1%) and 9 (25.7%) patients had mild, moderate, and severe disease respectively (Table 1).

NIHSS at presentation was similar in the two groups, with a median of 5.5 (IQR; 3, 12) and 4 (IQR; 3, 8) for the SARS-CoV-2 and control groups, respectively, P = 0.711. There was no statistically significant difference in the occluded vessel for large vessel occlusion (LVO) cases (S1 Table). Rate of IV thrombolytic treatment (IVT) was lower in the SARS-CoV-2 group with only 3 patients (8.6%), compared to 8 patients (22.9%) in the control group, and endovascular therapy was higher in the SARS-CoV-2 group with 9 patients (25.7%) compared to 4 patients (11.1%) in control group, however statistical significance was not reached Table 2.

CRP values were significantly higher in the SARS-CoV-2 group with a mean ($\pm$SD) of 4.5 mg/L ($\pm$5.9) compared to 1 mg/L ($\pm$1.6) in the control group, P = 0.003. There was no statistically significant difference regarding other laboratory values, including WBC, PLT and Fibrinogen Table 2.

**Table 1. Demographic and clinical characteristics.**

|  | SARS-CoV-2 n = 35 | Control n = 35 | P value |
|---|---|---|---|
| Female, n(%) | 14 (40) | 14 (40) | 1 |
| Age, median (IQR) | 74 (69,81) | 74 (68,82) | 0.810 |
| HTN, n(%) | 24 (68.6) | 27 (77.1) | 0.420 |
| AF, n(%) | 9 (25.7) | 7 (20.0) | 0.569 |
| DM, n(%) | 22 (62.9) | 16 (45.7) | 0.150 |
| Dyslipidemia, n(%) | 19 (54.3) | 23 (65.7) | 0.329 |
| Smoking, n(%) | 7 (20) | 7 (20.0) | 1.000 |
| IHD, n(%) | 9 (25.7) | 12 (34.3) | 0.434 |
| Prior TIA/Stroke, n(%) | 10 (28.6) | 11 (31.4) | 0.794 |
| Neoplasm, n(%) | 4 (11.8) | 7 (20.0) | 0.350 |
| **SARS-CoV-2 severity**: |  |  |  |
| Asymptomatic | 10 (28.6) |  |  |
| Mild | 10 (28.6) |  |  |
| Moderate | 6 (17.1) |  |  |
| Severe | 9 (25.7) |  |  |

HTN, hypertension; AF, atrial fibrillation; IHD, ischemic heart disease; TIA, transient ischemic attack

**Table 2. Stroke characteristics.**

|  | SARS-CoV-2 | control | p-value |
|---|---|---|---|
| NIHSS- admission, median (IQR) | 5.5 (3,12) | 4 (3,8) | 0.711 |
| **Etiology—TOAST, n(%)** |  |  |  |
| L | 8 (22.9) | 10 (28.6) | 0.485 |
| C | 11 (31.4) | 9 (25.7) |  |
| S | 9 (25.7) | 12 (34.3) |  |
| U | 7 (20.0) | 3 (8.6) |  |
| O | 0 (0.0) | 1 (2.9) |  |
| **Treatment, n(%)** |  |  |  |
| IVT | 3 (8.6) | 8 (22.9) | 0.101 |
| EVT | 9 (25.7) | 4 (11.4) | 0.124 |
| **Laboratory on admission, mean (SD)** |  |  |  |
| WBC (K/ul) | 9.1 (4.1) | 8.3 (2.9) | 0.346 |
| PLT (K/ul) | 273.9 (110.7) | 261 (93.5) | 0.623 |
| CRP (mg/L) | 4.5 (5.9) | 1 (1.6) | 0.003 |
| Fibrinogen (mg/dL) | 596.7 (162.4) | 549 (117.9) | 0.231 |

TOAST: L, large-vessel atherosclerosis; C, cardioembolic; S small-vessel occlusion; U, stroke of undetermined etiology; O, other determined etiology

IVT, intravenous thrombolytic treatment; EVT, endovascular treatment

## Stroke etiology and outcomes

There was no statistically significant difference between the groups in terms of stroke etiology but a trend toward a lower proportion of small vessel occlusion was seen in the SARS-CoV-2 group with 9 patients (25.7%) compared to 12 patients (34.3%) in control group, and a higher proportion of undetermined etiology with 7 patients (20%) compared to 3 patients (8.6%) in control group Table 2.

Regarding clinical outcomes, patients in the SARS-CoV-2 group had a lower rate of functional independence at day 90 (mRS≤2) with 13 (41.9%) patients compared to 17 (53.1%) patients in the control group, higher median mRS of 3 (IQR;1, 6) compared to 2 (IQR; 1, 4), and a higher mortality rate at day 90 of 11 patients (31.4%) compared to 6 (17.1%) in control group, but statistical significance was not reached in neither outcome (S2 Table).

Logistic regression analysis (adjusted for age, IVT and EVT) for the effect of SARS-CoV-2 infection on poor clinical outcome (defined as a higher mRS) also did not reach statistical significance [OR 1.675, 95% CI, 0.567–4.944, P = 0.350] (S3 Table). However, a logistic regression analysis within the SARS-CoV-2 group, found that a more severe SARS-CoV-2 disease predicted a higher mRS score at day 90 [OR 2.524, 95% CI, 1.166–5.465, P = 0.019]. Age and CRP values were not found to be significant predictors for higher mRS score within the SARS-CoV-2 group Table 3.

## Neurovascular imaging characteristics

Upon review of the CTA scans of patients in the SARS-CoV-2 group, we encountered an unusual pattern of floating arterial mural thrombi in 5/35 patients (14.3%). In four of these patients, all elderly males with cardiovascular risk factors, the floating thrombus was attached to a stenotic atherosclerotic plaque in the origin of the internal carotid artery, and it was ipsilateral to a middle cerebral artery (MCA) territory infarction. In two patients the carotid

**Table 3. Predictors for higher mRS score in SARS-CoV-2 patients.**

| | OR | 95% C.I. | | p-value |
|---|---|---|---|---|
| | | Lower | Upper | |
| Age | 1.06 | 0.975 | 1.151 | 0.172 |
| CRP | 0.947 | 0.828 | 1.083 | 0.426 |
| SARS-CoV-2 severity | 2.525 | 1.166 | 5.465 | 0.019 |

multivariant ordinal regression within SARS-CoV-2 group

stenosis was severe and in the other two it was moderate. The stenotic plaque was ulcerated in three of these four patients. Three patients received dual antiplatelet treatment for a period of up to three mounts and had successful carotid artery stenting performed (one patient in the acute phase of the stroke and the other two in the 2–3 weeks that followed). At one year, all three patients were alive, and no additional ischemic strokes were reported. The fourth patient only received dual antiplatelet treatment and died while in the hospital due to sepsis with aspiration pneumonia.

The 5th patient was a 69-year-old female which suffered an anterior circulation embolic stroke without evidence of carotid artery atherosclerosis. A cardiac CTA that was performed in search of an embolic origin showed multiple large mural thrombi in the descending aorta, with signs of only mild underlying atherosclerosis. This patient was treated with anticoagulation which resulted in full resolution of the thrombi. Anticoagulation was stopped after 5 months, and no further ischemic events were reported over one year of follow-up. All SARS-CoV-2 patients with floating thrombi are detailed in Table 4. Representative images of floating thrombi in SARS-CoV-2 patients are shown in Figs 1 and 2.

In the control group, a floating mural thrombus was detected in only one patient (2.9%) with a severe atherosclerotic RICA stenosis (P = 0.088).

Comparing the patients with SARS-CoV-2-associated floating mural arterial thrombi to the rest of the SARS-CoV-2 patients demonstrated a higher proportion of males (80% versus 66.7%), lower rates of atrial fibrillation and ischemic heart disease, an older age [median 78 years (IQR; 72.5,80) versus 73.5 (IQR; 64,81.75)] and a longer time from SARS-CoV-2 diagnosis to stroke onset [median 7 days (IQR; 0,15) versus 0 days (IQR; -0.5,8)]. No statistical significance was achieved for this analysis (S4 Table).

## Discussion

Initial reports on the risk of ischemic stroke in patients with SARS-CoV-2 infection were as high as 5.7% with severe infection [10]. Later studies reported lower estimates of 0.5%-1.6 [2, 11–13], which were still significantly higher when compared to patients hospitalized with other respiratory viruses [14].

The exact mechanism of thrombosis in patients with SARS-CoV-2 infection is not completely understood, but theoretically a cytokine storm that activates coagulation factors and inhibits fibrinolysis may contribute to the hypercoagulable state; Direct viral invasion of human cells through angiotensin-converting enzyme 2 (ACE2) receptors may lead to inflammatory endothelial damage [15, 16]; Derangements in coagulation parameters in critically ill patients including elevated D-dimer and fibrinogen levels, increased Factor VIII activity, low normal functional protein C activity, and increased von Willebrand factor activity have been reported [17–20] and antiphospholipid antibodies have been associated with SARS-CoV-2

**Table 4. Characteristics and imaging findings of patients with SARS-CoV-2-related stroke and floating mural arterial thrombi.**

| Age and Sex | Past medical history | SARS-Cov-2 severity | Time from SARS-Cov-2 diagnosis to stroke (days) | Stroke presentation | Neurovascular Imaging findings | Treatment | Post-stroke follow-up |
|---|---|---|---|---|---|---|---|
| 76, M | DM, HTN, prior TIA | Moderate | 7 | NIHSS 12 (rt. Hemiplegia, mild aphasia) | **A floating mural thrombus** on top of an ulcerated atherosclerotic plaque causing severe (90%) stenosis in the origin of the LICA | -Dual antiplatelet (3 months), followed by single antiplatelet. -CAS | mRS at day 90–3 mRS 1 year– 2 No additional reported strokes (2 years follow-up) |
| 69, F | HTN, HL, prior TIA | Mild | 13 | NIHSS 14 (rt. Hemiplegia, moderate aphasia) | **Multiple large mural thrombi in the descending aorta**, with signs of mild underlying atherosclerosis without significant stenosis. | - Acutely with tPA and M1 thrombectomy - Anticoagulation (5 months) - Antiplatelet | mRS at day 90–1 mRS 1 year– 1 No additional reported strokes (2 years follow-up) Imaging follow-up: Resolution of the thrombi on CTA after 3 months of Anticoagulation. |
| 79, M | DM | Severe | *0 | NIHSS 18 (rt. Hemiplegia, global aphasia) | **A floating mural thrombus** on top of an atherosclerotic plaque causing severe (95%) stenosis in the origin of the LICA | - Acutely with tPA and M1 thrombectomy - CAS - Antiplatelet | mRS at day 90–4 mRS 1 year– 4 No additional reported strokes (2.5 years follow-up) |
| 81, M | DM, HTN | Mild | *0 | NIHSS 4 (rt. Hand paresis, minor aphasia) | **A floating mural thrombus** on top of an ulcerated atherosclerotic plaque causing moderate (67%) stenosis in the origin of the LICA | - Dual antiplatelet (3 weeks), followed by single antiplatelet. - CAS | mRS at day 90–1 mRS 1 year– 1 No additional reported strokes (1 year follow-up) Imaging follow-up: Complete resolution of the thrombus with a remaining stenotic plaque after 4 days of DAPT. |
| 78, M | DM, HTN | Mild | 17 | NIHSS 20 (rt. Hemiplegia, global aphasia, forced eye deviation) | **A floating mural thrombus** on top of a large atherosclerotic plaque causing mild-to-moderate (50%) atherosclerotic stenosis in the origin of the LICA | - Dual antiplatelet - Anticoagulation | The patient died 1m after stroke (mRS-6) |

M, male; F, female; HTN Hypertension; DM diabetes mellitus; HL, hyperlipidemia, NIHSS, National Institutes of Health Stroke Scale; LICA, the left internal carotid artery; CAS, carotid artery stenting; mRS, modified Ranking score.

*0 SARS-Cov-2 and stroke diagnosis at the same time.

infection [21]. Hospital immobilization, ICU stay, use of vasopressors and hypoxia are additional factors that increase the risk of thrombosis.

Arterial thrombosis (AT), although reportedly less common than venous thromboembolism (VTE) in SARS-CoV-2 patients, may have dire consequences. The incidence of AT in SARS-CoV-2 patients may be as high as 15% with a mortality rate of up to 40%. Most patients reported are male and elderly with comorbidities and the most common location is limb arteries with a significant proportion of amputation or palliative care. When present, AT is reported to cause symptomatic thromboembolism in most patients [22–24].

In the current study, we retrospectively explored the neurovascular imaging data of SARS-CoV-2-related versus matched non-SARS-CoV-2-related ischemic stroke patients. In 4/35 (11.4%) SARS-CoV-2-related stroke patients we found an internal carotid artery (ICA) mural floating thrombus. All four patients had an ipsilateral embolic anterior circulation stroke and negative evaluation for cardioembolic stroke. Consequently, the ICA mural thrombus, via artery-to-artery embolism, represents the most probable etiological origin for stroke in these patients. In another SARS-CoV-2-related stroke patient, imaging showed multiple large mural

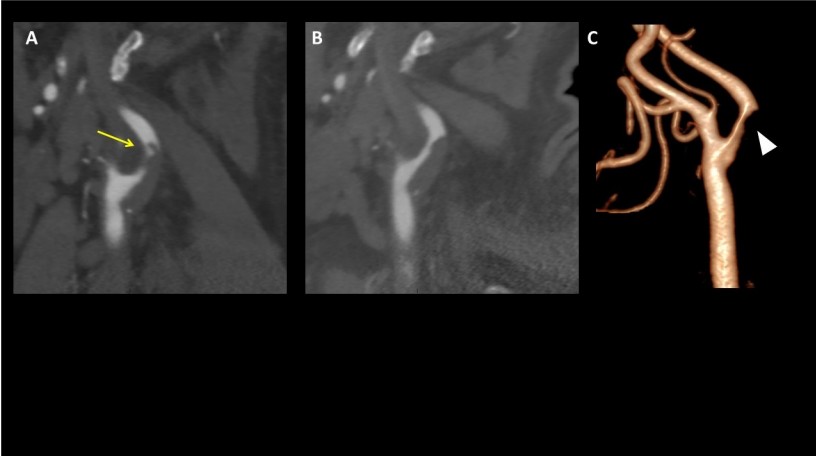

**Fig 1. CTA of an 81-year-old patient with SARS-CoV-2-related stroke that was attributed to large vessel atherosclerosis. A**. CTA on admission demonstrates a moderate ulcerated atherosclerotic stenosis of the LICA at the bifurcation with a floating thrombus (yellow arrow). **B**. CTA after 4 days of treatment with dual antiplatelet and a prophylactic dose of low molecular weight heparin demonstrates complete dissolution of the thrombus with a remaining stenotic plaque. C. 3D reconstruction of the left Carotid bifurcation, on admission.

thrombi in the descending aorta. Although direct artery-to-artery cerebral embolism from the descending aortic thrombosis is unlikely, the mural thrombi represent systemic hypercoagulability with a tendency to arterial thrombosis as the most probable stroke etiology in this patient as well. In the non-SARS-CoV-2-related ischemic stroke group a mural thrombus was found in only 1/35 (2.9%) patient (P = 0.088). A literature review of case reports on carotid artery mural floating thrombus in non-SARS-CoV-2 patients similarly found a very low prevalence (1.6%) of this imaging finding. This study also reported a male predominance (2:1) and a high proportion (47%) of hypercoagulability based on either active cancer or positive serological testing in these patients [25]. These data suggest that large-artery mural thrombosis with distal

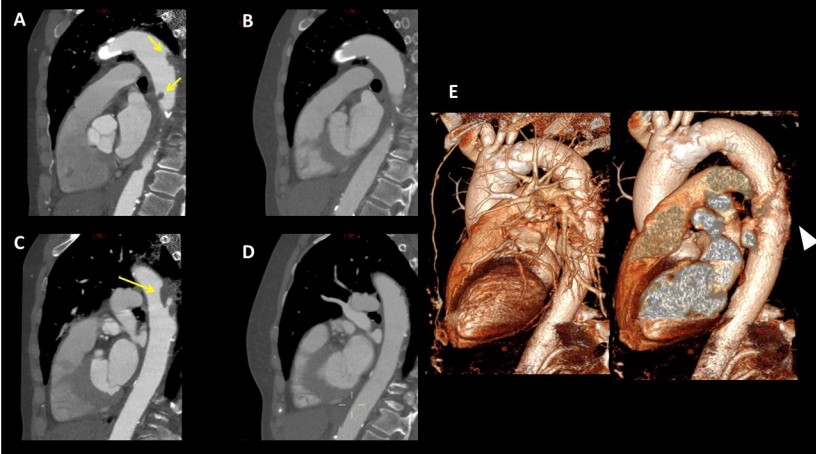

**Fig 2. CTA of a 69-year-old patient with SARS-CoV-2-related stroke attributed to an embolic source. A** and **C**: CTA on admission demonstrates multiple large thrombi in the descending aorta (yellow arrows) with only mild underlying atherosclerosis. **B** and **D**: CTA after 3 months of treatment with full anticoagulation shows complete resolution of the thrombi. E. 3D reconstruction of the descending Aorta, on admission.

embolization may be a typical stroke mechanism in SARS-CoV-2-related stroke whereas it seems to be exceptionally rare in the general stroke population.

The clinical severity of the infectious disease in SARS-CoV-2 patients emerged as an important risk factor for ischemic stroke in previous studies [10, 14, 26, 27]. However, a large body of evidence has accumulated on the occurrence of large vessel arterial thrombosis and ischemic stroke in patients with mild or asymptomatic SARS-CoV-2 infection [28–32]. In accordance with these data, 20/35 (57%) of patients with SARS-CoV-2-related stroke in our study had either mild or asymptomatic disease. Only 9/35 (26%) had severe disease. Among the five patients with mural floating thrombi, three patients had mild disease and only one had severe disease.

When compared to other patients with SARS-CoV-2-related stroke, patients with floating mural thrombosis in our study were more often males (80%), had a longer time from SARS-CoV-2 diagnosis to stroke onset (median 7 days), had more diabetes and hypertension and less atrial fibrillation and ischemic heart disease. Mural thrombi were situated on top of atherosclerotic stenosing plaques in all four patients with ICA thrombosis, but stenosis was only moderate in two of these patients. In the patient with descending aortic mural thrombi, there was only mild, flat underlying atherosclerosis. Our findings suggest that the formation of large artery mural thrombosis in SARS-CoV-2-related stroke may occur on otherwise benign-appearing, low-risk atherosclerotic plaques. This supports the notion that mild carotid atherosclerosis may be particularly prone to thrombus formation because of the unique combination of endothelitis and SARS-CoV-2-associated coagulopathy [33]. However, other studies are needed to clarify the exact underlying pathogenic mechanism. Notably, a recent pathological analysis of cerebral thrombi retrieved by endovascular thrombectomy in patients with SARS-CoV-2-associated stroke suggested neutrophils as the possible culprit in SARS-CoV-2-related thrombosis [34].

Previous studies reported the maximal risk of ischemic stroke during the first 3 days after SARS-CoV-2 diagnosis [35] with a median time interval from SARS-CoV-2 diagnosis to ischemic stroke of 2–3 days [11, 35]. Similarly, in our study the mean interval from SARS-CoV-2 diagnosis to ischemic stroke was 0 (-0.5,8) days for patients with SARS-CoV-2-related stroke without a mural thrombus. The median time interval for the 5 patients with a mural thrombus, however, was longer at 7 (0,15) days. A similarly prolonged median interval of 10.5 days was recently described for this subgroup of patients [21]. This prolonged interval may simply reflect the latency between asymptomatic mural thrombus formation and distal embolization resulting in ischemic stroke, or possibly arterial thrombosis overlying atherosclerosis in SARS-CoV-2-related patients may require a longer time to develop in comparison to cardioembolic or lacunar stroke. This finding requires confirmation in a larger cohort and further investigation.

A recently published case report and literature review identified 16 patients with symptomatic free-floating carotid thrombus in relation to a SARS-CoV-2 infection. The results strongly corroborate our findings: median time from COVID-19 diagnosis to the onset of neurologic complications was 10.5 days, the majority of patients (69%) were men and vascular risk factors were present in 70% of patients; in most patients CTA revealed non-occlusive thrombus without or with mild atherosclerosis plaque and in two patients a thrombus in the aortic arch was also found; all patients presented with stroke as the chief symptom and SARS-CoV-2 manifestations of mild severity [25].

Unlike previous reports of sporadic cases described in the literature, our retrospective case-control study included all patients with SARS-CoV-2-related stroke admitted to one medical center during the COVID-19 pandemic and compared them to matched non-SARS-CoV-2-related stroke patients admitted at a parallel period. Consequently, we are the first to report

mural floating thrombi as a unique, frequent, and significant stroke etiology in patients with SARS-CoV-2-related stroke in contrast to other stroke patients. Based on these findings, we recommend that CTA or MRA of the head and neck performed in patients diagnosed with SARS-CoV-2 and embolic acute ischemic stroke should be carefully inspected for arterial mural thrombosis in the carotid, aortic or other major cranio-cervical vessels as a potential artery-to-artery embolic source.

Literature presenting the treatment and prognosis of mural floating thrombi in non-SARS-CoV-2 patients contains low-quality evidence. These lesions reportedly pose a high short-term risk of stroke and death, but there is no definite answer for treatment in terms of anticoagulation or timing of intervention [36]. Intraluminal thrombi in the cervico-cephalic arteries were reported to have favorable clinical in-hospital course with low stroke recurrence, high rate of thrombus resolution, and good functional outcome when treated with combination antithrombotic, i.e. heparin with a single antiplatelet agent [37]. The prognosis of SARS-CoV-2-related stroke with mural thrombus in our study was likewise favorable with a higher proportion of good functional outcome at 90 days and lower mortality in comparison to other SARS-CoV-2-related stroke patients. Three of our patients with SARS-CoV-2-related stroke were treated with dual antiplatelet therapy. In all these patients, mural thrombus completely resolved within several days, carotid stenting was performed successfully, and no recurrent strokes were reported during a 2-year follow-up. These data suggest that dual antiplatelet therapy, which is the treatment of choice in patients with symptomatic carotid stenosis, may also suffice in cases of mural thrombosis to protect patients from recurrent stroke and cause dissolution of the thrombus.

Our study has several limitations. It is a single-center retrospective study, and a relatively low number of patients were included in the analysis. Our choice of using historical controls may have introduced bias related to unmeasured confounders such as improvements in stroke management over time, changes in medication, or lifestyle factors. Functional outcomes were assessed based on the medical file or phone calls by an unblinded physician, which may have introduced bias. Some of the SARS-CoV-2-related stroke patients were assigned a stroke etiology according to the TOAST classification based on incomplete stroke evaluation due to technical difficulties with performing ancillary tests during the COVID-19 pandemic.

**In conclusion**, arterial floating mural thrombus attached to an atherosclerotic plaque is a common and characteristic imaging finding in SARS-CoV-2-related stroke patients. Floating mural thrombi may form on otherwise low-risk plaques in patients with mild or asymptomatic infection and result in ischemic stroke up to 1–2 weeks after SARS-CoV-2 diagnosis. Patients diagnosed with embolic acute ischemic stroke and a recent SARS-CoV-2 diagnosis should perform a CTA or MRA of the head and neck to evaluate for arterial mural thrombosis in the craniocervical vessels as a potential embolic source.

## Supporting information

**S1 Table. Vascular anatomy for LVO patients.**
(DOCX)

**S2 Table. Stroke outcomes.**
(DOCX)

**S3 Table. Predictors of higher mRS score.**
(DOCX)

**S4 Table. Sub-analysis of SARS-CoV-2 patient's characteristics.**
(DOCX)

## Author Contributions

**Conceptualization:** Keshet Pardo, Omer Harnof, Shlomi Peretz.

**Data curation:** Keshet Pardo, Omer Harnof, Rani Barnea, Jonathan Naftali, Gilad Kenan, Shlomi Peretz.

**Formal analysis:** Keshet Pardo.

**Supervision:** Eithan Auriel.

**Writing – original draft:** Keshet Pardo, Shlomi Peretz.

**Writing – review & editing:** Keshet Pardo, Eithan Auriel, Shlomi Peretz.

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
