## [Decision Letter · Decision Letter 0]

19 Aug 2024

PONE-D-24-16018Arterial floating mural thrombi are a characteristic imaging pattern in SARS-CoV-2-related ischemic strokePLOS ONE

Dear Dr. Pardo,

Thank you for submitting your manuscript to PLOS ONE. After careful consideration, we feel that it has merit but does not fully meet PLOS ONE’s publication criteria as it currently stands. Therefore, we invite you to submit a revised version of the manuscript that addresses the points raised during the review process.

Both of the reviewers evaluated your study, so please respond the promlems indicated by them.

We look forward to receiving your revised manuscript.

Kind regards,

Yoshiaki Taniyama, MD, PhD

Academic Editor

PLOS ONE

2. Please amend either the abstract on the online submission form (via Edit Submission) or the abstract in the manuscript so that they are identical.

Reviewers' comments:

Reviewer's Responses to Questions

**Comments to the Author**

1. Is the manuscript technically sound, and do the data support the conclusions?

Reviewer #1: Yes

Reviewer #2: Yes

2. Has the statistical analysis been performed appropriately and rigorously? 

Reviewer #1: No

Reviewer #2: Yes

3. Have the authors made all data underlying the findings in their manuscript fully available?

Reviewer #1: Yes

Reviewer #2: Yes

4. Is the manuscript presented in an intelligible fashion and written in standard English?

Reviewer #1: Yes

Reviewer #2: Yes

5. Review Comments to the Author

Reviewer #1: This article presents a study on ischemic stroke in patients with SARSCoV2 acute infection compared to a control group created by matching for age and gender historical cases (pre covid-19 pandemic). The article is overall well written and provides interesting descriptive data that shed a light on possible pathogenetic mechanisms of stroke in SARSCoV2 patients. However, the use of a control group made of historical controls introduces a potential risk of bias that should be clearly recognized in the limitation section. Moreover other minor revisions are required:

1) Lines 61 to 66 “In addition to the common symptoms of fever, rhinorrhea, cough, and shortness of breath, SARS-CoV-2 may cause a hypercoagulable state associated with acute inflammatory changes. The risk for venous thromboembolism (VTE) is markedly increased. The risk of arterial thrombotic events such as stroke, myocardial infarction, and limb ischemia are also increased, but to a lesser extent than venous thrombosis. Stroke has been reported in 0.5-1.6 percent of hospitalized SARS-CoV-2 patients. Risk factors for arterial thrombosis included older age, male sex, history of coronary artery disease and D-dimer >230 ng/mL on presentation”

could you please provide references for these concepts? If they refer to reference 2, I think the D-dimer threshold for significantly increased risk of thrombotic events is >500. Please verify and correct.

2) lines 94-97: “The control group comprised of non-SARS-CoV-2 associated stroke patients from a pre- COVID-19 pandemic period, matched for gender and age (±5 years) with a case to control ratio of 1:1. The control patients were searched for during a period that predated the COVID-19 pandemic in Israel - between January 2018 and February 2020.”

As previously said, please add a short paragraph in the limitation section to acknowledge the risk of bias related to the choice of using historical controls (risk of introduction of unmeasured confounders such as improvements in stroke management over time, changes in medication, or lifestyle factors).

3) line 256 you wrote “infective disease”. Please correct into “infectious disease”

4) line 276-279: you wrote: “Similarly, in our study the mean interval from SARS-CoV-2 diagnosis to ischemic stroke was 3.3 (±6.7) for patients with SARS-CoV-2-related stroke without a mural thrombus. The mean time interval for the 5 patients with a mural thrombus, however, was longer at 7 (±7.64) days.”

The time interval from SARS-CoV-2 diagnosis to ischemic stroke does not seem to follow a normal distribution. I suggest to report it as median with interquartile range. Moreover, I suggest not to introduce new data in the discussion but to introduce them in the results section first.

Reviewer #2: a small interesting study on SARS-CoV2 related stroke:

please revise following aspects:

- the number of patients reported is rather low considering the center that receives about 1000 cases per year and the recruitment period of about 2 years - please comment;

-please provide 3d recosnstruction of all cases provided to enhance the visibility of the thrombus;

- is a dynamic imaging (Echo) available to see the floating characteristics of the thrombi? this would be important;

- please provide accurate quantification of carotid arterty plaques in all patients even if not stenosing;

- I would dampen this statement as up to know only case reports are reported and also the present work is a case series line 253-255 „These data suggest that large- artery mural thrombosis with distal embolization is a typical stroke mechanism in SARS- CoV-2-related stroke whereas it is exceptionally rare in the general stroke population“

- line 271-273 please add that however other studies are needed to calrify the exact underlying pathogenic mechanism; in this view please report a recent study that showed also a different composition of Sars-Cov2 related thrombi (10.1186/s40478-022-01313-y);

- please comment also the possibility of mural thrombi in non SARS-CoV2 positivie patients? what might be there predisoposiing conditions?

- line 299-303- authors should reword the sentence stating that particular attentions should be paid by neuroradiologist to analyse images to exclude possible mura thrombi on CTA imaging. In fact CTA neck vessel imaging should be always performed in stroke patients as soon as possible.

6. PLOS authors have the option to publish the peer review history of their article (what does this mean?). If published, this will include your full peer review and any attached files.

Reviewer #1: **Yes: **Domenico Spinelli

Reviewer #2: No

---

## [Author Response · Author response to Decision Letter 0]

28 Aug 2024

Response to reviewers

Reviewer #1: This article presents a study on ischemic stroke in patients with SARSCoV2 acute infection compared to a control group created by matching for age and gender historical cases (pre covid-19 pandemic). The article is overall well written and provides interesting descriptive data that shed a light on possible pathogenetic mechanisms of stroke in SARSCoV2 patients. However, the use of a control group made of historical controls introduces a potential risk of bias that should be clearly recognized in the limitation section. Moreover other minor revisions are required:

1) Lines 61 to 66 “In addition to the common symptoms of fever, rhinorrhea, cough, and shortness of breath, SARS-CoV-2 may cause a hypercoagulable state associated with acute inflammatory changes. The risk for venous thromboembolism (VTE) is markedly increased. The risk of arterial thrombotic events such as stroke, myocardial infarction, and limb ischemia are also increased, but to a lesser extent than venous thrombosis. Stroke has been reported in 0.5-1.6 percent of hospitalized SARS-CoV-2 patients. Risk factors for arterial thrombosis included older age, male sex, history of coronary artery disease and D-dimer >230 ng/mL on presentation”

could you please provide references for these concepts? If they refer to reference 2, I think the D-dimer threshold for significantly increased risk of thrombotic events is >500. Please verify and correct. Data were cited from reference 2. Indeed, the D-dimer threshold for significant increased risk of thrombotic events was >500. We corrected this number in introduction line 66.

2) lines 94-97: “The control group comprised of non-SARS-CoV-2 associated stroke patients from a pre- COVID-19 pandemic period, matched for gender and age (±5 years) with a case to control ratio of 1:1. The control patients were searched for during a period that predated the COVID-19 pandemic in Israel - between January 2018 and February 2020.”

As previously said, please add a short paragraph in the limitation section to acknowledge the risk of bias related to the choice of using historical controls (risk of introduction of unmeasured confounders such as improvements in stroke management over time, changes in medication, or lifestyle factors). Excellent point. We added this comment in the limitation section page 12 lines 322-324.

3) line 256 you wrote “infective disease”. Please correct into “infectious disease” corrected

4) line 276-279: you wrote: “Similarly, in our study the mean interval from SARS-CoV-2 diagnosis to ischemic stroke was 3.3 (±6.7) for patients with SARS-CoV-2-related stroke without a mural thrombus. The mean time interval for the 5 patients with a mural thrombus, however, was longer at 7 (±7.64) days.”

The time interval from SARS-CoV-2 diagnosis to ischemic stroke does not seem to follow a normal distribution. I suggest to report it as median with interquartile range. Moreover, I suggest not to introduce new data in the discussion but to introduce them in the results section first. Excellent point. We now reported these data in the form of median with interquartile range. Please notice that these data already appeared in the end of the results section line 218-219 ("..and a longer time from SARS-CoV-2 diagnosis to stroke onset [mean 7.4 days (SD, 7.6) versus 3.4 days (SD, 6.7)].") , before being mentioned again in the discussion section (now lines 288-290). We corrected to median in both places.

Reviewer #2: a small interesting study on SARS-CoV2 related stroke:

please revise following aspects:

- the number of patients reported is rather low considering the center that receives about 1000 cases per year and the recruitment period of about 2 years - please comment; In our study we found 35 SARS-CoV2 related stroke patients among ~2,000 stroke patients hospitalized during the same time period. This prevalence of 1.75% is comparable to that reported in other studies. See for example BMJ Open. 2024 Mar 28;14(3):e081527 stating a similarly low prevalence of 0.6%

-please provide 3d recosnstruction of all cases provided to enhance the visibility of the thrombus; Thank you for this excellent idea. 3D reconstruction now provided in figures 1+2.

- is a dynamic imaging (Echo) available to see the floating characteristics of the thrombi? this would be important; Unfortunately, dynamic imaging is not performed routinely in our center in this indication and is therefore not available.

- please provide accurate quantification of carotid arterty plaques in all patients even if not stenosing; accurate quantification of carotid artery stenosis now provided in table 4 for all patients with mural thrombus and atherosclerotic disease.

- I would dampen this statement as up to know only case reports are reported and also the present work is a case series line 253-255 „These data suggest that large- artery mural thrombosis with distal embolization is a typical stroke mechanism in SARS- CoV-2-related stroke whereas it is exceptionally rare in the general stroke population“ We rephrased this statement in Page 9, lines 259-260.

- line 271-273 please add that however other studies are needed to calrify the exact underlying pathogenic mechanism; in this view please report a recent study that showed also a different composition of Sars-Cov2 related thrombi (10.1186/s40478-022-01313-y); We now added this phrase and referred to the above study (lines 279-283). 

- please comment also the possibility of mural thrombi in non SARS-CoV2 positive patients? what might be their predisoposing conditions? Excellent point. In our study only one patient in the non SARS-CoV2 patients had a mural thrombus. We added a few sentences citing a literature review of case reports on carotid artery mural floating thrombus in non-SARS-CoV-2 patients that similarly found a very low prevalence of this imaging finding and reported risk factors. Page 9 lines 254-258.

- line 299-303- authors should reword the sentence stating that particular attention should be paid by neuroradiologists to analyze images to exclude possible mural thrombi on CTA imaging. In fact CTA neck vessel imaging should be always performed in stroke patients as soon as possible. Good point. We rephrased this sentence, now emphasizing that CTA performed in these patients should be carefully inspected for mural thrombi. Page 11 lines 311-313.

---

## [Decision Letter · Decision Letter 1]

17 Sep 2024

Arterial floating mural thrombi are a characteristic imaging pattern in SARS-CoV-2-related ischemic stroke

PONE-D-24-16018R1

Dear Dr. Pardo,

We’re pleased to inform you that your manuscript has been judged scientifically suitable for publication and will be formally accepted for publication once it meets all outstanding technical requirements.

Kind regards,

Yoshiaki Taniyama, MD, PhD

Academic Editor

PLOS ONE

Additional Editor Comments (optional):

Reviewers' comments:

Reviewer's Responses to Questions

**Comments to the Author**

1. If the authors have adequately addressed your comments raised in a previous round of review and you feel that this manuscript is now acceptable for publication, you may indicate that here to bypass the “Comments to the Author” section, enter your conflict of interest statement in the “Confidential to Editor” section, and submit your "Accept" recommendation.

Reviewer #2: All comments have been addressed

2. Is the manuscript technically sound, and do the data support the conclusions?

Reviewer #2: Yes

3. Has the statistical analysis been performed appropriately and rigorously? 

Reviewer #2: Yes

4. Have the authors made all data underlying the findings in their manuscript fully available?

Reviewer #2: Yes

5. Is the manuscript presented in an intelligible fashion and written in standard English?

Reviewer #2: Yes

6. Review Comments to the Author

Reviewer #2: (No Response)

7. PLOS authors have the option to publish the peer review history of their article (what does this mean?). If published, this will include your full peer review and any attached files.

Reviewer #2: No

---

## [Editor Report · Acceptance letter]

24 Sep 2024

PONE-D-24-16018R1 

PLOS ONE

Dear Dr. Pardo, 

I'm pleased to inform you that your manuscript has been deemed suitable for publication in PLOS ONE. Congratulations! Your manuscript is now being handed over to our production team.

Kind regards, 

on behalf of

Dr. Yoshiaki Taniyama 

Academic Editor

PLOS ONE